

# The nutritional status of children living within institutionalized care: a systematic review

Emily DeLacey[1,2,3], Cally Tann[3,4,5,6], Nora Groce[7], Maria Kett[7], Michael Quiring[2], Ethan Bergman[8], Caryl Garcia[2] and Marko Kerac[1,3]

[1] Department of Population Health, Faculty of Epidemiology and Population Health, London School of Hygiene & Tropical Medicine, University of London, London, United Kingdom
[2] Holt International, Eugene, OR, United States of America
[3] Centre for Maternal, Adolescent, Reproductive, & Child Health (MARCH), London School of Hygiene & Tropical Medicine, University of London, London, United Kingdom
[4] Department of Infectious Disease Epidemiology, Faculty of Epidemiology & Population Health, London School of Hygiene & Tropical Medicine, University of London, London, United Kingdom
[5] MRC/UVRI & LSHTM Uganda Research Unit, London School of Hygiene & Tropical Medicine, University of London, Entebbe, Uganda
[6] Neonatal Medicine, University College London Hospitals NHS Trust, London, United Kingdom
[7] UCL International Disability Research Centre, Department of Epidemiology and Health Care, University College London, London, United Kingdom
[8] Department of Health Sciences, College of Education and Professional Studies, Central Washington University, Ellensburg, WA, United States of America

Corresponding author
Emily DeLacey,
emilyd@holtinternational.org

## ABSTRACT

**Background**. There are an estimated 2.7 million children living within institutionalized care worldwide. This review aimed to evaluate currently available data on the nutrition status of children living within institutionalized care.

**Methods**. We searched four databases (Pubmed/Medline, CINHAL Plus, Embase and Global Health Database) for relevant articles published from January 1990 to January 2019. Studies that included information on anthropometry or micronutrient status of children living within institutionalized care were eligible for inclusion. The review is registered on PROSPERO: CRD42019117103.

**Results**. From 3,602 titles screened, we reviewed 98 full texts, of which 25 papers were eligible. Two (8%) studies reported data from multiple countries, nine (36%) were from Asia, four (16%) from Africa, three (12%) from Eastern Europe, four (16%) from the European Union and one (4%) from each of the remaining regions (Middle East, South America and the Caribbean). Twenty-two (88%) were cross sectional. Ten (40%) of the studies focused on children >5 years, seven (28%) on children <5 years, seven (28%) covered a wide age range and one did not include ages. Low birth weight prevalence ranged from 25–39%. Only five (20%) included information on children with disabilities and reported prevalence from 8–75%. Prevalence of undernutrition varied between ages, sites and countries: stunting ranged from 9–72%; wasting from 0–27%; underweight from 7–79%; low BMI from 5–27%. Overweight/obesity ranged from 10–32% and small head circumference from 17–41%. The prevalence of HIV was from 2–23% and anemia from 3–90%. Skin conditions or infections ranged from 10–31% and parasites from 6–76%. Half the studies with dietary information found inadequate intake or diet diversity. Younger children were typically more malnourished than older children, with a few exceptions. Children living within institutions were more

malnourished than community peers, although children living in communities were also often below growth standards. High risk of bias was found.

**Conclusions**. This study highlights the limited amount of evidence-based data available on the nutritional status of children in institutions. Of the studies reviewed, children living within institutionalized care were commonly malnourished, with undernutrition affecting young children particularly. Micronutrient deficiencies and obesity were also prevalent. Data quality was often poor: as well as suboptimal reporting of anthropometry, few looked for or described disabilities, despite disability being common in this population and having a large potential impact on nutrition status. Taken together, these findings suggest a need for greater focus on improving nutrition for younger children in institutions, especially those with disabilities. More information is needed about the nutritional status of the millions of children living within institutionalized care to fully address their right and need for healthy development.

birth weight

# INTRODUCTION

Malnutrition impacts millions of children around the world (*Black et al., 2013*; *The World Bank Group, 2019*; *UNICEF, 2019*). In 2018 for children younger than 5 years old, 49 million children were wasted, just under one in four (21.9%) were stunted and 5.9% were overweight (*UNICEF, 2019*). Almost half of the deaths among children younger than 5 years old have undernutrition as an underlying factor (*Black et al., 2013*; *UNICEF, 2019*). In some countries, up to half of adolescents are stunted, as many as 11% are too thin, up to 5% are obese and over 50% are anemic (*Black et al., 2013*). Being malnourished has many adverse consequences including increased risk and severity of infections, increased risk of disability, and death (*Black et al., 2013*; *Groce et al., 2014*; *McDonald et al., 2013*; *Myatt et al., 2018*). This can be a part of a cyclical interaction between infections and undernutrition which leads to poor nutritional status, illnesses and impacted growth. The first 1,000 days of a child's life are particularly important because poor nutrition at this stage also predisposes children to long-term impairments such as stunted growth, impaired cognition and poor performance at school and work (*Black et al., 2013*; *UNICEF, 2019*).

Some children are at higher risk of malnutrition, such as orphans and children living within institutionalized care (*UNICEF, 2019*). UNICEF estimates that there are some 140 million orphans worldwide who have lost either one or both of their parents (*UNICEF, 2017*). Although most orphans live with other family members, some live in institutionalized care or residential care facilities (*UNICEF, 2017*). Institutionalized care is defined by the United Nations as residential care that is provided in any non-family-based group setting, including all other short- and long-term residential care facilities (*United Nations General Assembly, 2009*). Many non-orphans live in institutionalized care for a variety of reasons, including social or economic (*van IJzendoorn et al., 2011*; *The Children's Health Care Collaborative Study Group, 1994*). These children are also vulnerable (*Baron, Baron*

*& Spencer, 2001*; *The Children's Health Care Collaborative Study Group, 1994*). Though family-based care is the ideal environment for all children, this is not always possible (*Petrowski, Cappa & Gross, 2017*; *The Children's Health Care Collaborative Study Group, 1994*).

Approximately 2.7 million children ages 17 years and younger live in residential care globally: 120 children per 100,000 (*Petrowski, Cappa & Gross, 2017*). The UN Convention on the Rights of the Child states that when it is in a child's best interest and they cannot remain in their family, alternative-care options need to be provided for the child. Alternative-care solutions include foster care or institutional care. These alternative-care options need to meet a standard of living adequate for a child's full development, including children with disabilities; particularly in regard to education, health, development, nutrition and other essentials (*United Nations Human Rights Office of the High Commissioner, 1990*).

Children in institutional care often face numerous adversities prior to admission and many enter institutionalized care with pre-existing nutritional, developmental, medical and neurological conditions (*Baron, Baron & Spencer, 2001*; *The Children's Health Care Collaborative Study Group, 1994*; *The St Petersburg-USA Orphanage Research Team, 2005*; *The St. Petersburg- USA Orphanage Research Team, 2008*). Some have disabilities or were born prematurely or with low birth weight and many have had exposure to drugs or alcohol, HIV, stress or a range of other issues—all of which can impact their health. (*Baron, Baron & Spencer, 2001*; *Groce et al., 2014*; *The Children's Health Care Collaborative Study Group, 1994*; *The St. Petersburg- USA Orphanage Research Team, 2008*). Often there is limited or no information about children's early lives or exposures prior to coming into care (*The Children's Health Care Collaborative Study Group, 1994*; *The St Petersburg-USA Orphanage Research Team, 2005*; *The St. Petersburg- USA Orphanage Research Team, 2008*). Those entering institutionalized care may experience further negative issues when admitted: ongoing risk of suboptimal nutrition, poor growth or growth failure, neglect or abuse, impacted physical and mental development, diarrhea, anemia, infections and diseases because of the conditions in the care centers (*Frank et al., 1996*; *Johnson & Gunnar, 2011*; *The Children's Health Care Collaborative Study Group, 1994*). Disability can be both a contributing factor and a result of malnutrition. In addition, disabilities, micronutrient deficiencies and malnutrition can all lead to increased morbidities and mortality (*Groce et al., 2014*; *McDonald et al., 2013*; *Myatt et al., 2018*).

Often because of limited staffing, time and fiscal constraints, institutions are able to only provide basic care needs for children instead of addressing children's individual needs for healthy and full development (*van IJzendoorn et al., 2011*; *Whetten et al., 2014*). Factors impacting children's nutrition status in care centers include inadequate or poor quality of food or inappropriate types of food; inadequate stimulation or attention; improper use of medications; inappropriate feeding practices; and poor hygiene and sanitation leading to frequent illnesses and negatively impacting utilization of nutrients (*Frank et al., 1996*; *van IJzendoorn et al., 2011*; *The St. Petersburg- USA Orphanage Research Team, 2008*).

## METHODS

The aim of our review was to better understand the current nutritional status of children in care by looking at anthropometric and nutritional status indicators in relation to age, disability, geography, gender and related factors, with an ultimate goal of improving policy and practice to better meet the needs of this unique and vulnerable population.

We analyzed existing published peer-reviewed literature on the nutrition status of children in institutional care by examining anthropometric data, micronutrient status and other factors including disability status, gender and age. PRISMA guidelines were followed throughout the review process and a PROSPERO registration was completed prior to the start of the study (PROSPERO 2019: CRD42019117103, https://www.crd.york.ac.uk/PROSPERO/display_record.php?RecordID=117103) (*Moher et al., 2009*; *National Institute for Health Research, 2019*).

The review primarily evaluated observational and intervention studies. Inclusion criteria included material published between January 1990 and January 2019 in English and contained research related to orphanages/institutionalized care, children, nutrition, anthropometric data or micronutrient status. We selected these dates because the Convention on the Rights of the Child went into effect in 1990, and since then, there have been significant changes in institutional care and changes in the understanding of the needs of children in institution-based care (IBC) (*Frank et al., 1996*; United Nations Human Rights Office of the High Commissioner).

In order to be included in this review, the studies must have addressed a population of children younger than 18 years old (with the exception of one study which included children as old as 20 years but was retained for informational value), been peer reviewed and included at least one measurement of nutrition status through standardized tools, such as WHO Growth Standards or WHO Growth References and definitions (*World Health Organization, 2019a*; *World Health Organization, 2019b*). Anthropometric indicators of interest included: weight for age, length/height for age, weight for length/height, head circumference for age and mid-upper arm circumference for age. Micronutrient status, clinical signs/symptoms and dietary information were also included when available. Emily DeLacey, the principal investigator, and Dr. Marko Kerac determined and used the search strategy. Four electronic databases were searched through OVID from December 2018 through January 2019: Pubmed/Medline, CINHAL Plus, Embase and Global Health Database. For details of our search strategy, see Appendix S1. Initial article screening was based on title and abstract, following which full texts were assessed for eligibility against our pre-specified inclusion/exclusion criteria. Discussions with the research team resolved any questions of eligibility with Dr. Cally Tann deciding any discords. A data extraction table was used to summarize key information from the final selection of articles into tables and columns organized by related themes and areas.

Nutritional status was determined according to reported anthropometry, whether reported by $z$-scores (standard deviations from a reference population) or percentiles. Micronutrient status and intake were also reported on and included prevalence of anemia or micronutrient deficiencies. Other key data areas included disability status, birth weight,

sex, age, dietary intake and any reported disease, illness or infection which could impact nutrition status. Heterogeneity in the type of interventions prevented our ability to conduct a meta-analysis of the study, so a narrative synthesis was used.

## RESULTS

We found a total of 3,973 papers. After 371 duplicates were removed, the remaining 3,602, were screened by title and abstract. All but 98 articles were excluded during this phase. Of the 98 identified as potentially eligible, we were unable to locate seven, 53 had insufficient anthropometry or used non-standard measurements, 10 did not have appropriate population or study type and three were excluded because the anthropometric data existed in another study. Twenty-five studies met our inclusion criteria (Fig. 1). Most studies (22, 88%) were cross sectional (Table 1). The most commonly researched region was Asia with nine studies (36%), followed by Africa with four studies (16%). Three (12%) were from Eastern Europe, four (16%) from the European Union and one (4%) from each of the remaining regions (Middle East, South America and the Caribbean). Kenya and India were the most commonly researched countries and were each included in four studies (Table 1). Ten (40%) focused on children older than 5 years, seven (28%) on children younger than 5 years, seven (28%) covered a wide age range and one did not include ages. Twelve (48%) included control or comparison groups of children who were community children (CC) or orphaned, separated or abandoned children living in family-based care (FBC), or children living on the streets (CLS). Control groups were typically orphaned children living in family-based care (FBC) or community children (CC) with no history of institutional care and the groups were selected from different settings including from local schools, communities, clinics or hospitals, lists, house-to-house census or other child-related programs (*Braitstein et al., 2013*; *Johnson et al., 2010*; *Whetten et al., 2014*). Eight (32%) studies mentioned or analyzed gender differences (Tables 1 and 2). A history of low birth weight (LBW) were also common (25% to 39%).

### Children with disabilities

Of the 25 studies reviewed, 12 (48%) did not state whether they included children with disabilities (Tables 1 and 2). Eight (32%) of the studies stated that children with disabilities were excluded, leaving only five (20%) mentioning children with disabilities in their reporting, but either excluded them from analysis or did not state whether or not they were excluded. Only one study included any anthropometric measurements for children with disabilities (*Lewindon et al., 1997*). The St. Petersburg-USA Orphanage Research Team found that 21% of children had disabilities (*The St Petersburg-USA Orphanage Research Team, 2005*). Miller and colleagues found 16% of institution-based children (IBC) had significant disabilities and 75% had developmental delays (*Miller et al., 2006*).

### Anthropometrics

Undernutrition, micronutrient deficiencies and overweight/obesity were reported in varying ways. Prevalence of undernutrition differed markedly: stunting (low length/height for age) from 9 to 72%; wasting (low weight for length/height) from 0 to 27%; underweight
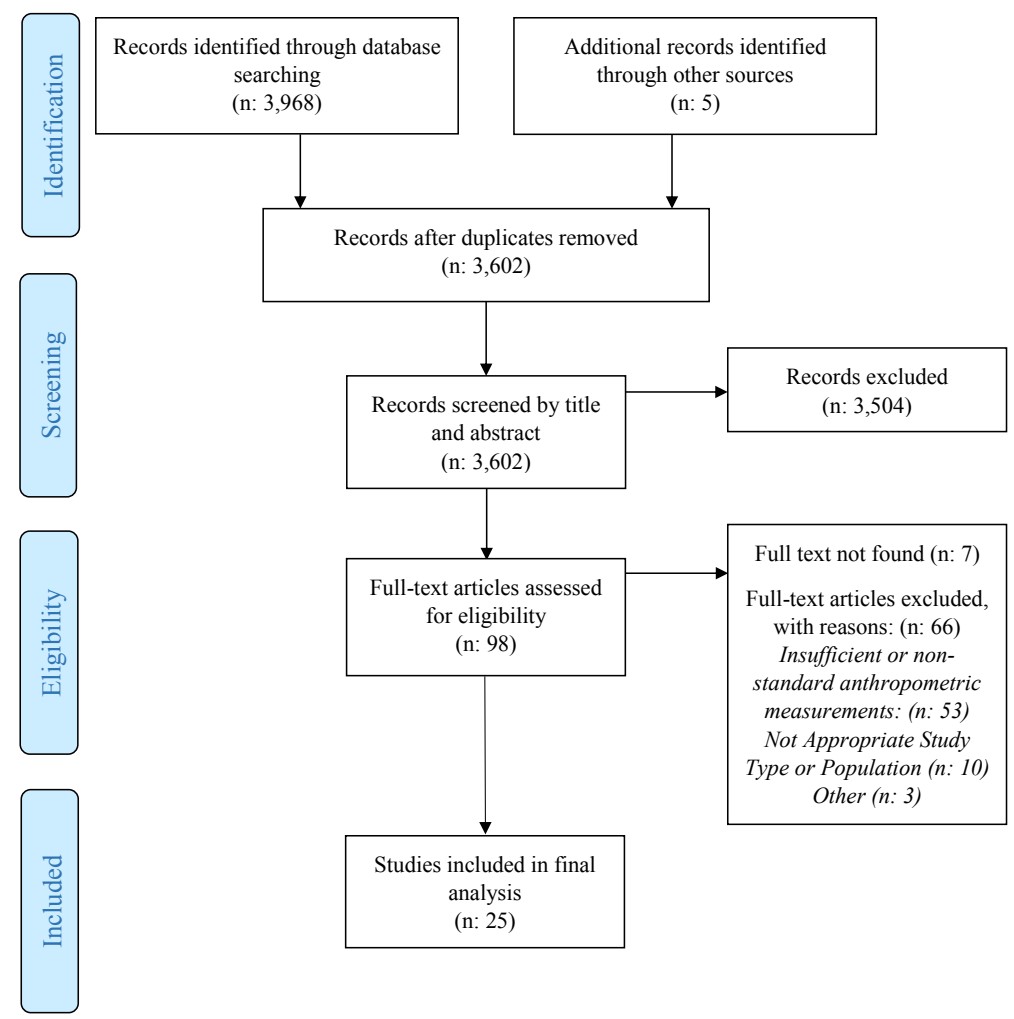

**Figure 1** **PRISMA flow diagram.**

(low weight for age) from 7 to 79%; low BMI (body mass index) ranged from 5 to 27% (Table 2). Ten to 32% of children were overweight or obese. Panpanich et al. found children younger than 5 years old to be more stunted, wasted and underweight than older children and below WHO growth standards (*Panpanich et al., 1999*). The prevalence of small head circumference ranged from 17 to 41%.

## Micronutrients, clinical signs/symptoms and infections

Clinical signs or symptoms were reported in 48% (12) of the studies (Table 3). Five (20%) mentioned HIV but two of these were conducted in institutions for children with HIV (*Kapavarapu et al., 2012*; *Myint et al., 2012*). Excluding the facilities for children with HIV, HIV prevalence was from 2 to 23%. One study found a higher prevalence of morbidity among children in IBC than CC ($p < 0.05$) (*Mwaniki, Makokha & Muttunga, 2014*). The prevalence of parasites ranged from 6 and 76%, with Lesho and colleagues finding 10%

**Table 1  Description of studies included in review.** Description of studies included in the review of children living within institutionalized care.

| Author, year | Study design | Country | Number of institutions | Study population | Gender (percent female) | Disability |
|---|---|---|---|---|---|---|
| | | | Multi-Country | | | |
| Whetten et al. (2014) | Longitudinal Cohort | Cambodia, Ethiopia, India, Kenya, Tanzania | 83 | n: 2,283, IBC: 993 (43.5%) and FBC: 1,290 (56.5%), median age 9 years at baseline, range 6–12 years and median age 12 years at year 3 follow-up, range 8–16 years | IBC: 43%, FBC: 47% | Unknown, Special needs homes excluded |
| Whetten et al. (2009) | Cross Sectional | Cambodia, Ethiopia, India, Kenya, Tanzania | 83 | n: 2,837, IBC: 1,480, 6–12 years, mean age 9 years, FBC: 1,357 | IBC: 42.8%, FBC: 47.1% | Unknown, Special needs homes excluded |
| | | | Africa | | | |
| Aboud et al. (1991) | Cross Sectional | Ethiopia | 1 | n: 81, 5–14 years, IBC mean age 9.5 years ± 2.8, FBC mean age 9.7 ± 2.6 | 25.9 % | Unknown |
| Braitstein et al. (2013) | Cross Sectional | Kenya | 19 | n: 2862, IBC: 1337, FBC: 1425, CLS: 100, 0–18 years, median age 11.1 years | 46% | Unknown, HIV included |
| Mwaniki, Makokha & Muttunga (2014) | Cross Sectional | Kenya | 4 Schools (multiple orphanages attended) | n: 416, IBC: 208, CC:208, range 4–11 years, 50% 4–7 years and 50% 8–11 years | 50% | Excluded |
| Panpanich et al. (1999) | Cross Sectional | Malawi | 3 | n: 293, IBC: 76, mean age 6.44 ± 4.69, range 0-<15 years, FBC: 137, mean age 7.92 ± 2.62, CC: 80, mean age 6.1 ± 3.17 | Total: 45.4% , IBC: 44.7%, FBC: 44.5%, CC: 47.4% | Unknown, HIV included |

DeLacey et al. (2020), *PeerJ*, DOI 10.7717/peerj.8484

**Table 1** (*continued*)

| Author, year | Study design | Country | Number of institutions | Study population | Gender (percent female) | Disability |
|---|---|---|---|---|---|---|
| | | | | Asia | | |
| *Bin Shaziman et al. (2017)* | Cross Sectional | Malaysia | 5 | n: 85, 13–18 years | – | Excluded |
| *Chowdhury et al. (2017)* | Cross Sectional | Bangladesh | 1 | n: 232, 6–18 years, mean age 13.38 years ± 3.69 | 44% | Excluded |
| *Hearst et al. (2014)* | Cross Sectional | Kazakhstan | 10 | n: 308 children, 0- 3 years | – | Excluded |
| *Kapavarapu et al. (2012)* | Prospective Longitudinal | India | 1 | n: 85, mean age 9.2 years, range 4–14 | 40% | Unknown, HIV group home |
| *Kroupina et al. (2014)* | Cross Sectional | Kazakhstan | 6 | n: 103, ages 5–29 months, mean 14.89 months ± 6.85) | 49.5% | Excluded |
| *Lewindon et al. (1997)* | Cross Sectional | Hong Kong | 1 | n: 215, 11.9 years ± 5.2, range 1.9–27 | 47% | Included, 3 residential wards for children with disabilities |
| *Myint et al. (2012)* | Cross Sectional | Myanmar | 1 | n: 60, 2–15 years, >5: 26.7%, 5–10: 56.7%, 11–15: 16.6% | 53.3% | Unknown, HIV group home |
| *Sarma et al. (1991)* | Cross Sectional | India | 70 | 3,822, 6–18 years | – | Unknown |
| *Zahid & Karim (2013)* | Cross Sectional | Bangladesh | 1 | n: 49, 6–15 years, mean age 8.72 years ± 1.38 | 61% | Included, 8.7% |

DeLacey et al. (2020), *PeerJ*, DOI 10.7717/peerj.8484

**Table 1** (*continued*)

| Author, year | Study design | Country | Number of institutions | Study population | Gender (percent female) | Disability |
|---|---|---|---|---|---|---|
| | | | Eastern Europe | | | |
| *Lesho et al. (2002)* | Cross Sectional | Moldova | – | n: 367 | – | Unknown |
| *Miller et al. (2006)* | Cross Sectional | Russia | 3 | n: 234, mean age 21 months ± 12.6, range 1.5 months to 6 years | 45% (gender not recorded for 12 children) | Included, 16% severe disabilities, 75% developmental disabilities but excluded from analyses |
| *The St Petersburg-USA Orphanage Research Team (2005)* | Cross Sectional | Russia | 3 | n: 325 children, 0–5 years | – | Included, 8% of the intake sample (N: 383) but 21% of the children in residence (N:302) were considered to have a disability but excluded from analyses |
| | | | European Union | | | |
| *Johnson et al. (2010)* | Cross Sectional | Romania | 6 | n:136, mean age 21 months ± 7.32; range 5 months- 2.7 years | 50% | Excluded |
| *Martins et al. (2013)* | Prospective Longitudinal | Portugal | 15 | n: 49, mean 7.14 months ± 6.17) range 0–21 months | 49% | Excluded |
| *Pysz, Leszczynska & Kopec (2015)* | Cross Sectional | Poland | 5 | n:153, range 7–20 years | 43.8% | Unknown |
| *Smyke et al. (2007)* | Cross Sectional | Romania | 6 | n: 208, IBC: 123, CC: 66, 5 months– 2.6 years, mean age 20.65 months ± 7.26 | IBC: 50.4% CC: 53% | Excluded |
**Table 1** (*continued*)

| Author, year | Study design | Country | Number of institutions | Study population | Gender (percent female) | Disability |
|---|---|---|---|---|---|---|
| | | | Middle East | | | |
| *El-Kassas & Ziade (2017)* | Cross Sectional | Lebanon | 2 | n: 153, 5–14 years, mean age 8.86 ± 2.45 years | 62.7% | Unknown |
| | | | South America | | | |
| *Nunes et al. (1999)* | Cross Sectional | Brazil | 1 | n: 243, 1–15years | 30.3% | Included, HIV included |
| | | | The Caribbean | | | |
| *Nelson (2016)* | Cross Sectional | Jamaica | 3 | n: 226, IBC n: 113, 5–18 years, mean 10.66 ± 3.67 years, CC n: 103, mean 10.28 years ± 3.20 | IBC: 38.9%, CC: 58.3% | Unknown, HIV and other infectious diseases excluded |

**Notes.**

Study population: IBC, Institution-based care; FBC, Family-based care (orphaned or abandoned children in community settings); CC, Community children (non-orphans); CLS, Children living on the street.

of children in IBC having three or more parasites (*Lesho et al., 2002*). Skin infections, varicella zoster, tuberculosis, impetigo, dental issues, ear/nose/throat problems, respiratory infections, diarrhea and other conditions or illnesses were frequently reported among IBC (Table 3). Skin conditions or infections ranged between 10 and 31%, and Kapavarapu and colleagues found 75% of children had an infection within the first three months of admission to a site (*Kapavarapu et al., 2012*). Seven (28%) reported on micronutrient status or intake and the prevalence of anemia ranged from 3 to 90%. Hearst and colleagues found over a third of children had low vitamin D (*Hearst et al., 2014*). Other micronutrient deficiencies discussed included iodine, zinc, albumin, as well as vitamins A and B (Table 3). Edema, conjunctival pallor, xerophthalmia and goiters were found more in children in IBC than those living in FBC (*Aboud et al., 1991*).

## Dietary diversity, intake and food security

Eight (32%) studies discussed dietary diversity, intake or food security (Table 3). Mwaniki and colleagues found that diet diversity was lower in children living in IBC than for CC ($p < 0.05$). Diets were reported to have a high reliance on starches and legumes (*Mwaniki, Makokha & Muttunga, 2014*). Of the studies that assessed dietary intake, 50% found adequate intake. Dietary adequacy varied; from children in IBC at 3.9 times higher risk of consuming inadequate calories to having 362% higher intake than estimated average requirements for some nutrients. The one study which reported on food security found that children in IBC had higher food security when compared to children in FBC, 42% vs. 2% (*Braitstein et al., 2013*).

## DISCUSSION

The nutritional status of children living in institutions has the potential to adversely impact their health and well-being, yet out of 3,602 papers from four major databases, only 25 peer-reviewed papers presented evidence based findings on the children's nutrition status (Fig. 1). All 25 reviewed studies indicated that many of the children in institutionalized care faced some form of malnutrition. The available data suggests that children living within institutionalized care are commonly malnourished: affected by undernutrition, overweight and micronutrient deficiencies. With few exceptions, mostly of older children, children living within institutionalized care were significantly below standards for growth, diet and micronutrient status and were often below comparison groups of their community peers. Nutrition status varied between care centers and between the ages of children, with younger children at a higher risk of being malnourished. There may be a number of reasons why this is the case, such as younger children have a harder time feeding themselves, especially if disabilities are present, and young, poorly nourished children are at risk of not surviving to become older children in institutional settings (*McDonald et al., 2013*; *Myatt et al., 2018*; *The Children's Health Care Collaborative Study Group, 1994*). Diet inadequacy, micronutrient deficiencies and illnesses or infections were also found to be prevalent in children of all ages.

To our knowledge, this is the first systematic review of the nutrition status of children living within institutionalized care. It is important because 2.7 million children worldwide

**Table 2 Anthropometric measurements and results.** Anthropometric data of children living within institutionalized care in various countries.

| Author, year | Growth reference | Weight for age (WAZ) | Weight for-length/ height (WHZ) | Length/ height for age (HAZ) | BMI-for-Age | Head circumference for age | Other | Results |
|---|---|---|---|---|---|---|---|---|
| | | | | Multi-Country | | | | |
| *Whetten et al. (2014)* | WHO growth charts | – | – | IBC: Mean −1.0 ± 1.4, FBC: Mean −1.0 ± 1.3 | IBC: Mean −0.7 ± 1.0, FBC: Mean −0.7 ± 1.2 | – | – | This study does not support the hypothesis that IBC is systematically associated with poorer well-being than FBC for orphaned and abandoned children ages 6 to 12 in countries with high rates. Much greater variability among children within care settings was observed than among care-setting types. |
| *Whetten et al. (2009)* | WHO growth charts | – | – | IBC: Mean −0.96 ± 1.46, FBC: Mean −1.03 ± 1.29, Weighted IBC vs. FBC: Mean (CI) 0.011 (−0.08, 0.10) | IBC: Mean −0.68 ± 0.97, FBC: Mean −0.73 ± 1.39, Weighted IBC vs. FBC: Mean (CI) 0.072 (−0.01, 0.16) | – | – | While it is possible that respondent bias accounts for better subjective health scores for IBC, the lack of significant differences on the biometric scores and the lower prevalence of recent illness suggest that the growth and overall health of IBC is no worse than that of FBC. There were no differences between children in IBC and FBC in mean height for age or BMI for age. |
| | | | | Africa | | | | |
| *Aboud et al. (1991)* | NCHS | IBC: >80%: 64% <80%: 36% FBC: >80%: 73.5% <80%: 25.6% p = NS | IBC: >80%: 97.3% <80%: 2.7% FBC: >80%: 95.6% <80%: 4.1% p = NS | IBC: >90%: 76% <90%: 24% FBC: >90%: 91.8% <90%: 8.2% p < 0.05 | – | – | – | The children in IBC were more likely to be short for their age indicating early and chronic malnutrition. Both groups of children had a high probability of weighing less than the standard for their age. Using both anthropometric and clinical signs of malnutrition, 27 (33%) IBC showed nutritional problems on two or more indices. |
| *Braitstein et al. (2013)* | WHO | ≤ 10 years, n: 2131 ≥-2 z-scores OR unadjusted IBC: 1 FBC: 0.87 (0.56–1.34) | ≤ 5 years, n: 380 ≥-2 z-scores OR unadjusted IBC: 1 FBC: 1.02 (0.55–1.90) | 0–18 years, n: 2842 ≥-2 z-scores OR unadjusted IBC: 1 FBC: 2.27 (1.74–2.94) CLS: 4.95 (3.13–7.82) % Stunting IBC: 59% FBC: 74% CLS: 88% | 10–18 years, n: 2374 ≥-2 z-scores OR unadjusted IBC: 1 FBC: 0.70 (0.49–1.01) CLS: 0.58 (0.31–1.08) High BMI (p < 0.001) IBC: 10% FBC: 16% CLS: 19% | – | – | FBC were more than twice as likely as children in IBC to be stunted (AOR: 2.6, 95% CI [2.0–3.4]). CLS were nearly six times more likely to be stunted compared to children in IBC (AOR: 5.9, 95% CI [3.6–9.5]). IBC have improved nutrition status and are more likely to have an adequate diet and much less likely to be stunting compared to FBC. Children in IBC were more likely to be normal weight for height compared to FBC (p = 0.024) |
| *Mwaniki, Makokha & Muttunga (2014)* | *World Health Organization Multicentre Growth Reference Study Group (2006)* | IBC n: 69 % underweight: 33.2% CC n: 31 % underweight: 14.9% Total n: 100 % underweight: 24% p > 0.0001 | IBC n: 19 % wasted: 9.2% CC n: 20 % wasted: 9.7% Total n: 39 % wasted: 9.4% p = 0.866 | IBC n: 98 % stunted: 47.2% CC n: 51 % stunted: 24.5% Total n: 149 % stunted: 35.8% p > 0.0001 | – | – | – | The risk of stunting was 2.8 times higher and underweight was 0.043 times higher among IBC compared with CC. |

DeLacey et al. (2020), *PeerJ*, DOI 10.7717/peerj.8484

**Table 2** (*continued*)

| Author, year | Growth reference | Weight for age (WAZ) | Weight for-length/ height (WHZ) | Length/ height for age (HAZ) | BMI-for-Age | Head circumference for age | Other | Results |
|---|---|---|---|---|---|---|---|---|
| *Panpanich et al. (1999)* | NCHS | <5 years Mean z-scores: IBC: −2.17 ± 1.46 FBC: −1.82 ± 1.19 CC: −1.37 ± 1.28 Moderate underweight (<-2 z-scores) %: IBC: 54.8% FBC: 33.3% CC: 30% Severe underweight (<-3 z-scores) %: IBC: 38.7% FBC: 16.7% CC: 6.7% ≥5 years Mean z-scores: IBC: −0.91 ± 0.96 FBC: −1.11 ± 1.10 CC: −1.24 ± 1.00 Moderate underweight (<-2 z-scores) %: IBC: 6.8% FBC: 23.9% CC: 20.8% | <5 years Mean z-scores: IBC: −0.35 ± 1.15 FBC: −0.68 ± 1.10 CC: −0.45 ± 0.93 Wasting (<-2 z-scores) %: IBC: 9% FBC: 12% CC: 0% ≥5 years Mean z-scores: IBC: −0.08 ± 0.91 FBC: −0.64 ± 0.99 CC: −0.53 ± 0.79 $p < 0.05$ for variance between the three groups Wasting (<-2 z-scores) %: IBC: 0% FBC: 5.3% CC: 2.3% | <5 years Mean z-scores: IBC: −2.75 ± 1.29 FBC: −2.20 ± 1.51 CC: −1.61 ± 1.57 $p < 0.05$ for variance between the three groups Stunting (<-2 z-scores) %: IBC: 64.5% FBC: 50% CC: 46.4% ≥5 years Mean z-scores: IBC: −1.07 ± 1.51 FBC: −1.07 ± 1.51 CC: −1.41 ± 1.41 Stunting(<-2 z-scores) %: IBC: 9.1% FBC: 30.4% CC: 34% | – | – | – | Younger than 5 years old, the mean z-scores of W/A, W/H and H/A for all groups were much lower than those of the NCHS reference population. More malnutrition of children in IBC younger than 5 years than those in FBC and CC. Girls were more malnourished in IBC than boys ($p < 0.05$). 44.1% IBC who stayed less than 1 year were undernourished compared with 12.2% who stayed ≥1 year ($p < 0.05$). Children in IBC ≥ 5 years of age were less stunted and wasted than FBC and CC, which suggests that children in IBC have greater long-term food security than FBC and CC. "Older orphanage children seem to have better nutrition than village orphans." |

|  |  |  |  | Asia |  |  |  |  |
|---|---|---|---|---|---|---|---|---|
| *Bin Shaziman et al. (2017)* | WHO Growth References | – | – | – | Severely thin 4.7% Thin 2.4% Normal 61.2% Overweight 16.5% Obesity 15.3% | – | – | – |
| *Chowdhury et al. (2017)* | WHO Growth References, Essence of Pediatrics 2011 ranges for malnutrition | Total malnourished: 60.3%, Mild: 43.1%, Moderate: 16.8%, Severe: 0.4% | – | – | – | – | – | Children 15 to 18 years old were most malnourished. Higher malnutrition among the boys than girls in the age group of 15–18 years old but gender did not have a significant effect on severity. Malnutrition was higher during the first four years in the orphanage. With increasing duration in the orphanage, malnutrition levels gradually declined. |
| *Hearst et al. (2014)* | *World Health Organization (1995), World Health Organization Multicentre Growth Reference Study Group (2006)* | n: 286, mean z-score: −1.3 ± 1.5, median −1.3 31.5% underweight | n: 286, mean z-score: −0.7 ± 1.5, median −0.6 22.1% wasting | n: 286, mean z-score: −1.5 ± 1.9, median −1.5 36.7% stunting | – | – | – | 72% of the children had one or more growth, nutrition or developmental deficits, and 24% had three or more deficits. The growth-related indicators coincide with the high prevalence of low albumin, indicating generalized chronic undernutrition and suggest macronutrient deficiencies that could be due to inadequate diets, infections and/or inflammation or impaired nutrient absorption or utilization secondary to the psychosocial stress of living in an institution. Prevalence for growth-related deficiencies and anemia in indicate IBC are more at risk compared with corresponding results for data from 90 CC of a similar age attending local child care centers. |

DeLacey et al. (2020), *PeerJ*, DOI 10.7717/peerj.8484

**Table 2** (*continued*)

| Author, year | Growth reference | Weight for age (WAZ) | Weight for-length/ height (WHZ) | Length/ height for age (HAZ) | BMI-for-Age | Head circumference for age | Other | Results |
|---|---|---|---|---|---|---|---|---|
| *Kapavarapu et al. (2012)* | NCHS, CDC, *World Health Organization Multicentre Growth Reference Study Group (2006)* | 25th percentile: −3.73 Median: −2.75 75th percentile: −2.05 Underweight (WAZ <−2): 79% Over 36 months median WAZ increased to −1.74, 25th percentile −2.46, 75th percentile-1.03 (*P* < 0.001). | 25th percentile: −2.29 Median: −1.30 75th percentile: −0.56 Wasting (WHZ <-2): 27% Median WHZ scores increased to −0.10, 25th percentile −0.18, 75th percentile −0.01 over 36 months (*P* = 0.49) | 25th percentile: −3.06 Median: −2.69 75th percentile: −1.94 Stunting (HAZ <-2): 72% Median HAZ also increased to −1.63, 25th percentile −2.19, 75th percentile: −0.77 (*P* < 0.001). | – | – | – | "Irrespective of the ART status, a decrease in under-weight, stunting and wasting was seen at the end of 36 months. There was an observed higher rate of *z*-score increase among children not yet on ART compared to that of those who were on ART was probably attributable to the fact that children on ART had a more advanced forms of disease along with co-morbidities which resulted in slower rate of improvement in growth than children with a milder form of disease and who did not need to be treated with ART. All received age and gender appropriate nutrition along with additional nutrition supplements such as iron when required. These results suggest that dietary support (both macronutrients and micronutrients) may have a role in improving nutritional outcomes in HIV-infected individuals, thereby improving quality of life and perhaps indirectly reducing disease-related mortality." |
| *Kroupina et al. (2014)* | *World Health Organization Multicentre Growth Reference Study Group (2006)* | Mean: −1.34 ± 1.17, range −4.9 to 0.94 <-2 *z*-scores: 22.3% | Mean: −0.63 ± 1.41, range −4.44 to 2.84 <-2 *z*-scores: 19.4% | Mean: −1.62 ± 1.61, range −5.49 to 3.11 <-2 *z*-scores: 35.5% | – | n:102, mean: −1.70 ± 1.27, range −4.53 to 1.90 <-2 *z*-scores: 41.2% | – | "We found that all three of the growth parameters departed substantially from expected levels relative to those of healthy children." Prevalence of low birth weight was 35%, compared to 6% national population, was found to be a significant negative predictor of developmental status. |
| *Lewindon et al. (1997)* | Not specified n:141 | Mean: −3.9 *z*-scores | – | – | – | – | Triceps Skin Fold Median: 58.6% | – |
| *Myint et al. (2012)* | WHO | – | – | Short Stature: 18.3% Stunted 45% | Underweight: 26.7% Overweight: 8.3% Obese: 1.7% | – | – | Nutritional problems seen in 60% of the children. "No significant difference in nutritional status nor proportion of short stature and stunted was seen among boys and girls. There is no association of HIV staging and nutritional status." |
| *Sarma et al. (1991)* | NCHS | Girls mean wt range (kg): 16.5 ± 2-46.8 ± 9.66 Boys mean wt range (kg): 16.3 ± 2.18-49.3 ± 6.96 | – | Girls mean ht range (cm): 104 ± 6.30-154.2 ± 5.64 Boys mean ht range (cm): 106 ± 6.52-166.0 ± 9.49 | – | – | Girls mean arm circumference (cm): 15 ± 0.78–22.7 ± 3.59 Boys mean arm circumference (cm): 14.5 ± 1.04-23.3 ± 0.60 | Growth was similar in all regions analyzed. Heights and weights were far below NCHS figures, suggesting a high degree of growth delay and stunting but were higher than urban slum or rural counterparts. The extent of delay, in terms of age, was up to 3 years. |

DeLacey et al. (2020), *PeerJ*, DOI 10.7717/peerj.8484

**Table 2** (*continued*)

| Author, year | Growth reference | Weight for age (WAZ) | Weight for-length/ height (WHZ) | Length/ height for age (HAZ) | BMI-for-Age | Head circumference for age | Other | Results |
|---|---|---|---|---|---|---|---|---|
| *Zahid & Karim (2013)* | Nutrition survey of Rural Bangladesh 1996 | Mean: −0.39 ± 1.22 Underweight: 13% Normal: 84.8% Overweight: 2.2% | Mean: 0.38 ± 1.36 Wasted: 2.7% Normal: 83.8%, Overweight: 8.1% Obese: 5.4% | Mean: −0.76 ± 1.02 Stunted: 8.7% Normal: 89.1% Tall: 2.2% | Underweight: 10.87% Normal: 60.87% Overweight: 21.74% Obese: 6.5% | – | – | – |
| | | | | | Eastern Europe | | | |
| *Miller et al. (2006)* | WHO (excluding head circumference which was compared to American standards) n: 201, mean z-scores (excluding CWD) | Birth: −1.34 ± 0.08 Placement: −1.59 ± 0.12 Present: −1.50 ± 0.12 | – | Birth: -.62 ± .14 Placement: −1.45 ± 0.13 Present: −1.48 ± 0.10 | – | Birth: −1.55 ± 0.12 Placement: −1.38 ± 0.11 Present: −1.20 ± 0.11 | – | 75% (84/112) of children's records available indicated developmental delays. Measurements did not differ significantly between boys and girls, nor did they correlate with age at placement or current age of the children. Children with a prior diagnosis of FAS tended to have lower anthropometric z-scores at all time points than those without this diagnosis, but the results were significant only for birth height ($p = 0.04$), birth weight ($p = 0.02$), and placement head circumference ($p = 0.01$). >90% of children with high phenotypic scores had moderate or severe developmental delays. |
| *The St Petersburg-USA Orphanage Research Team, 2005* | CDC, USA Vital Statistics, and standards for the Northwestern Region of the Russian Federation. | Mean: −1.68 (1.39) CC (n:66):- 0.06 (1.02) $p < 0.01$ Intake ($N = 327$, 309) Residents ($N = 236$, 216) Russian 10th percentile: 41–67% 25th percentile: 58–78% 50th percentile: 90–97% 75th percentile: 96–98% 90th percentile: 99% CDC 10th percentile: 55–63% 25th percentile: 73–81% 50th percentile: 90–91% 75th percentile: 97% 90th percentile: 99% | Mean: −0.60 (1.20) CC (n:66): 0.002 (0.99) Intake ($N = 294$, 304) Residents ($N = 231$, 219) Russian 10th percentile: 24% 25th percentile: 49–54% 50th percentile: 93–90% 75th percentile: 97–95% 90th percentile: 100–98% CDC 10th percentile: 29–25% 25th percentile: 49–50% 50th percentile: 93–90% 75th percentile: 97–95% 90th percentile: 100–98% | Mean: −1.56 (1.37) CC (n:60): 0.06 (0.98) $p < 0.001$ Intake ($N = 327$, 304) Residents ($N = 237$, 218) Russian 10th percentile: 34–54% 25th percentile: 49–73% 50th percentile:91–95% 75th percentile: 95–98% 90th percentile: 98–99% CDC 10th percentile: 43–61% 25th percentile: 61–77% 50th percentile: 78–90% 75th percentile: 93–96% 90th percentile: 97–99% | – | Mean: −1.17 (1.33) CC (n:60): 0.17 (0.79) p<0.001 Intake ($N = 329$, 298) Residents ($N = 238$, 197) Russian 10th percentile: 44–53% 25th percentile: 63–74% 50th percentile: 92–96% 75th percentile: 97–99% 90th percentile: 99–100% CDC 10th percentile: 44–46% 25th percentile: 64–68% 50th percentile: 89–85% 75th percentile: 97–91% 90th percentile: 98–97% | Chest Circumference Intake ($N = 329$) Residents ($N = 237$) Russian 10th percentile: 40–43% 25th percentile: 57–63% 50th percentile: 93–92% 75th percentile: 97–96% 90th percentile: 99% | Disabilities: prenatal narcotic exposure, fetal alcohol syndrome, physical deformity, Down syndrome, cerebral palsy, hydrocephalus, microcephalus, heart disorder, other. Non-Specific Disabilities: encephalopathy, growth insufficiency, dystrophy. HIV+ reside in a separate facility. Intake: 27% LBW, 5.5% VLBW Residents: 39.1% LBW, 8.8% VLBW For height, weight, head circumference and chest circumference, more than 35 to44% of the children at intake are below the 10th percentile for their gender in physical size relative to the northwestern Russian Federation and 43 to55% are below the 10th percentile of USA standards. Approximately 90% or more are below the median of both these standards. |
| | | | | | European Union | | | |
| *Johnson et al. (2010)* | CDC 2000 IBC: n:125, 21.0 months ± 7.4 CC: n: 72, 19.3 months ± 7.1 | IBC: mean −1.23 ± 1.08, P ≤.001 z-scores ≤−2: 25%, P ≤.001 CC: mean −0.05 ± 1.00 z-scores ≤−2: 0 | IBC: −0.67 ± 1.14, P ≤.001 z-scores ≤−2: 16%, P<.01 CC: 0.16 ± 0.96 z-scores ≤−2: 2% | IBC: mean −0.84 ± 0.86, P ≤.001 z-scores ≤−2: 9%, P<.05 CC: 0.13 ± 0.91 z-scores ≤−2: 2% | – | IBC: mean −1.10 ± 0.99, P ≤.001 z-scores ≤−2: 17%, P<.01 CC: −0.15 ± 0.86 z-scores ≤−2: 2% | – | 24% of children living in IBC compared to 3% CC were low birth weight (p ≤.001). |

DeLacey et al. (2020), *PeerJ*, DOI 10.7717/peerj.8484

**Table 2** (*continued*)

| Author, year | Growth reference | Weight for age (WAZ) | Weight for-length/ height (WHZ) | Length/ height for age (HAZ) | BMI-for-Age | Head circumference for age | Other | Results |
|---|---|---|---|---|---|---|---|---|
| *Martins et al. (2013)* | *World Health Organization (2009)*, Latent Class Analysis (LCA) Mean, SD | Persistently low (n: 10, 20.4%) Percentile T0 (admission): 1.23 ± 1.60 Percentile T1: 3.91 ± 6.52 Percentile T2: 2.11 ± 3.39 Percentile T3: 4.39 ± 6.07 Deteriorating (n: 12, 24.5%) Percentile T0 (admission): 19.04 ± 28.63 Percentile T1: 20.85 ± 23.25 Percentile T2: 15.48 ± 21.87 Percentile T3: 17.83 (18.47) Improving (n: 16, 32.7%) Percentile T0 (admission): 24.02 ± 26.42 Percentile T1: 27.92 ± 26.82 Percentile T2: 27.42 ± 28.85 Percentile T3: 30.13 ± 23.98 Persistently high (n: 11, 22.5%) Percentile T0 (admission): 59.45 (32.81) Percentile T1: 55.95 (27.71) Percentile T2: 52.71 (26.30) Percentile T3: 58.06 (28.73) | – | Persistently low (n: 18, 36.7%) Percentile T0 (admission): 3.17 ± 4.47 Percentile T1: 4.52 ± 5.24 Percentile T2: 2.32 ± 2.67 Percentile T3: 4.56 ± 4.39 Deteriorating (n: 9, 18.4%) Percentile T0 (admission): 44.51 ± 27.02 Percentile T1: 49.52 ± 12.37 Percentile T2: 21.44 ± 9.64 Percentile T3: 23.83 ± 15.70 Improving (n: 14, 28.6%) Percentile T0 (admission): 15.00 ± 10.00 Percentile T1: 18.17 ± 11.54 Percentile T2: 29.14 ± 26.88 Percentile T3: 32.47 ± 12.18 Persistently high (n: 8, 16.3%) Percentile T0 (admission): 76.41 ± 32.50 Percentile T1: 71.26 ± 29.18 Percentile T2: 72.82 ± 14.49 Percentile T3: 78.42 ± 20.35 | – | Persistently low (n: 11, 22.5%) Percentile T0 (admission): 5.92 ± 6.72 Percentile T1: 6.13 ± 6.35 Percentile T2: 10.05 ± 8.55 Percentile T3: 14.62 ± 13.79 Deteriorating (n: 9, 18.4%) Percentile T0 (admission): 34.43 ± 29.00 Percentile T1: 42.92 ± 29.14 Percentile T2: 37.79 ± 28.21 Percentile T3: 18.02 ± 14.35 Improving (n: 16, 32.7%) Percentile T0 (admission): 40.42 ± 26.75 Percentile T1: 55.36 ± 24.56 Percentile T2: 60.50 ± 12.84 Percentile T3: 66.05 ± 15.10 Persistently high (n: 13, 26.5%) Percentile T0 (admission): 68.93 ± 24.39 Percentile T1: 90.05 ± 8.58 Percentile T2: 89.58 ± 9.33 Percentile T3: 91.18 ± 7.89 | – | Being younger at institutional admission posed a significant risk factor for impaired physical development across the three domains. Being a boy was a risk factor for compromised growth in weight and head circumference. Findings lead the researchers to believe that slower growth rates may be linked to younger infants in depriving contexts being highly susceptible to insufficient stimulation and support. The data shows that the pre- and perinatal circumstances that precede institutionalization influence children's development in institutions. Children's physical status at birth was also significantly associated with their growth trajectories. Children born longer, heavier and with larger head circumferences stayed in the persistently high groups for height and weight. The most favorable weight trajectory was associated with better interactions with caregivers. |

DeLacey et al. (2020), *PeerJ*, DOI 10.7717/peerj.8484

**Table 2** (*continued*)

| Author, year | Growth reference | Weight for age (WAZ) | Weight for-length/ height (WHZ) | Length/ height for age (HAZ) | BMI-for-Age | Head circumference for age | Other | Results |
|---|---|---|---|---|---|---|---|---|
| *Pysz, Leszczynska & Kopec (2015)* | University of Physical Education in Krakow (percentiles) | – | – | – | Thinness or Underweight: 14% boys and 5% girls Normal BMI: 86% boys and 92% girls Overweight or obesity: 6% boys and 6% girls | – | Thickness of the sum of three skin folds in normal ranges: boys 83% and girls 85% | Thickness of skinfolds was measured in ∼90% of the participants both genders (in relation to a wide range of standards, between 10 and 90 percentiles). Strong correlation between the thickness of skinfold and gender. The average thicknesses of various skinfolds were higher in girls than in boys. |
| *Smyke et al. (2007)* | CDC IBC:123 CC: 62 | Mean $z$-scores: IBC: $-1.25 \pm 1.07$ CC: $-.06 \pm 1.02$ $p < 0.01$ | Mean $z$-scores: IBC: $-.79 \pm 1.03$ CC: $.002 \pm .99$ $p < 0.001$ | Mean $z$-scores: IBC: $-.89 \pm .90$ CC: $.06 \pm .98$ $p<0.001$ | – | Mean $z$-scores: IBC: $-.77 \pm .97$ FBC: $.17 \pm (.79)$ $p < 0.001$ | Size IBC: $-.93 \pm .77$ FBC: $.044 \pm (.89)$ $p < 0.001$ | Children living in IBC had poorer growth compared to CC. When birthweight was entered as a covariate, findings were similar, with the exception of weight for height, which was no longer significantly different. Physical size was examined and found that it was associated (positively) only with birth weight. |
| | | | | Middle East | | | | |
| *El-Kassas & Ziade (2017)* | *World Health Organization (2009)* | – | – | Stunting: <10 years: 11.3% ≥10 years: 16.4% Total: 13.7% $p = 0.352$ | Normal: 90.8% Overweight (≥+2SD): 7.2% Obese (≥+3SD): 2% $p=0.311$ | – | – | Increasing age (OR: 5.201, 95% CI [1.347–20.085]), irregular breakfast intake (OR: 6.852, 95% CI [1.462–32.12]), and increased screen time more than two hours per day (OR: 12.126, 95% CI [2.659–55.288]) were associated with significantly higher odds of being stunted. Older age group had a higher prevalence of overweight and obesity, compared to the younger age group. |
| | | | | South America | | | | |
| *Nunes et al. (1999)* | NCHS, type classified according to the Seone-Lathan classification | – | – | – | – | – | – | 41% were malnourished, including both chronic and acute malnutrition cases. 49% of the girls and 40% of the boys had malnutrition. No significant difference between malnourished children and controls. 3% cerebral palsy; 3% developmental delay; 2.1% with microcephaly; .8% with fetal alcohol syndrome; 4.3% ADDH; 1.3% Down syndrome. |

DeLacey et al. (2020), *PeerJ*, DOI 10.7717/peerj.8484

**Table 2** (*continued*)

| Author, year | Growth reference | Weight for age (WAZ) | Weight for-length/ height (WHZ) | Length/ height for age (HAZ) | BMI-for-Age | Head circumference for age | Other | Results |
|---|---|---|---|---|---|---|---|---|
| | | | | The Caribbean | | | | |
| Nelson (2016) | WHO | IBC Girls 5–11 years (n: 24): 0.006 ± 0.748 IBC Boys 5–11 years (n: 38): −0.229 ± 1.09 20% of IBC were mildly underweight, and 2.5% were moderately underweight. CC Girls 5–11 years (n: 39): 0.905 ± 1.30 CC Boys 5–11 years (n: 33): 0.252 ± 0.871 7.3% of CC were mildly underweight. | – | IBC Girls 5–11 years (n: 24): 0.509 ± 1.21 12–18 years (n: 20): 0.065 ± 0.962 IBC Boys 5–11 years (n: 33): −0.239 ± 1.29 12–18 years (n: 10): 0.991 ± 2.57 15.3% of IBC were mildly stunted, and 4.5% were moderately stunted. CC Girls 5–11 years (n: 39): 1.065 ± 0.984 12–18 years (n: 21): 0.785 ± 1.17 CC Boys 5–11 years (n: 33): 0.591 ± 0.928 12–18 years (n: 10): −0.044 ± 1.30 4.9% of CC were mildly stunted. | – | – | Mean MUAC IBC Girls 5–11 years (n: 24): 18.08 cm ± 2.0 12–18 years (n: 20) 22.55 cm ± 2.87 IBC Boys 5–11 years (n: 38): 17.35 cm ± 3.8 12–18 years (n:31): 23.21 cm ± 2.9 CC Girls 5–11 years (n: 39): 19.87 cm ± 3.6 12–18 years (n: 21) 24.01 cm ± 2.54 CC Boys 5–11 years (n: 33): 18.17 cm ± 2.1 12–18 years (n:10): 23.11 cm ± 3.1 Mean Triceps Skinfold IBC Girls 5–11 years (n: 39): 18.08 cm ± 2.0 12–18 years (n: 21) 22.55 cm ± 2.87 IBC Boys 5–11 years (n: 33): 17.35 cm ± 3.8 12–18 years (n:10): 23.21 cm ± 2.9 CC Girls 5–11 years (n: 39): 19.87 cm ± 3.6 12–18 years (n: 21) 24.01 cm ± 2.54 CC Boys 5–11 years (n: 33): 18.17 cm ± 2.1 12–18 years (n:10): 23.11 cm ± 3.1 | Children living in institutional care were at higher risk for malnutrition. Young girls living with family members had significantly better anthropometric assessments of growth as compared to their peers living in IBC. However, the effect sizes were small, explaining only 4.4% (HAZ) to 10.3% (WAZ) of the variance in measurements of nutritional status observed between these groups. |

**Notes.**

Study population: IBC, Institution-based Care; FBC, Family-based Care (orphaned or abandoned children in community settings); CC, Community Children (non-orphans); CLS, Children living on the Street.

WHO, World Health Organization; NCHS, National Center for Health Statistics (USA); CDC, Centers for Disease Control (USA); BMI, Body Mass Index; ht, height; wt, weight; MUAC, Mid-pper Arm Circumference.

DeLacey et al. (2020), *PeerJ*, DOI 10.7717/peerj.8484

**Table 3  Diet, micronutrient status, clinical signs/ symptoms and infections results.** Diet, micronutrient status, clinical signs/ symptoms and infections of children living within institutionalized care in various countries.

| Author, year | Dietary analysis | Micronutrient status | Clinical signs/symptoms and infections |
|---|---|---|---|
| | Multi-Country | | |
| Whetten et al. (2009) | – | – | By caregiver report, children living in institutions were also less likely to have had a cough, diarrhea or fever in the two weeks before the interview (19.9 vs. 41.2%, weighted difference 220.6%, 95% CI [224%,218%]) or to be sick on the day of the interview (5.9% vs. 12.2%,), weighted difference 26.1%, 95% CI [28%, 24%]). |
| | Africa | | |
| Aboud et al. (1991) | – | – | Edema: IBC (4%), FBC (0%) Conjunctival Pallor: IBC (4%), FBC (0%) Xerophthalmia: IBC (15.5%), FBC (8.2%) Goiter: IBC (2.7%), FBC (2%) Nutritional problems were not significantly more prevalent among IBC. |
| Braitstein et al. (2013) | Using the Household Food Insecurity Access Scale (HFIAS), 42% of IBC and 2% of FBC reported being food secure. 95% of children in IBC reported an adequate diet compared to 93% of children in FBC and 99% of SLC, ($p = 0.009$). | – | HIV rates: IBC (2.1%), FBC (1.3%), SLC (1%) ($p = 0.001$) |

**Table 3** (*continued*)

| Author, year | Dietary analysis | Micronutrient status | Clinical signs/symptoms and infections |
|---|---|---|---|
| *Mwaniki, Makokha & Muttunga (2014)* | Using a 24hr diet recall and Nutri Survey program, diets were assessed. A total of 63 and 37 food items were consumed by the CC and IBC respectively. Only 7.2% of IBC consumed more than three food groups compared to 45.2% of CC. 92.9% of IBC and 54.8% of CC consumed less than four food groups ($p < 0.05$). CC had significantly ($p < 0.05$) higher diversity of foods served than IBC. Energy intake: The total mean energy intake among CC was 1,890 Kcal per day and was significantly higher ($p < 0.05$) than that of IBC. The intake of energy by IBC who took lunch was 1,547 Kcal compared to the energy intake of CC who also took the three meals of the day ($p < 0.05$). The mean energy intake of IBC who did not take lunch was less than half of that of CC. IBC who attended school away from the orphanage had two meals (mainly breakfast and supper) in a day during school days and three meals during the weekend and did not meet their daily needs compared to CC who always had three meals. IBC had 3.9 times higher risk of consuming inadequate calories compared to CC. Orphanages tend toward exclusive reliance on starches and legumes. Food in orphanages mainly depended on donations. | – | IBC Diarrhea: 11.5% Cough/ colds: 12.5% Fever: 1.4% Vomiting/skin rashes: 7.7% FBC Diarrhea: 2.4% ($p = 0.015$) Cough/colds: 2.9% ($p = 0.14$) Fever: 0.5% ($p = 0.8$) Vomiting/skin rashes: 0.5% ($p = 0.006$) Prevalence of morbidity was significantly ($p < 0.05$) higher among the IBC compared to CC children and 1.2 times higher risk of being sick. IBC had significantly ($p < 0.05$) higher prevalence of diarrhea and cold/cough compared to CC. IBC were twice less likely to wash hands at critical times compared to CC. 48% of IBC reported washing hands after visiting the toilet the day before the interview compared to 78.2% of CC. 49.4% of IBC and 78.2% of CC washed their hands before meals. There was also a higher proportion (76.3%) of IBC who reported washing hands with soap during the critical times compared with the CC (12.8%) ($p < 0.01$). Vaccination among IBC compared to CC ($p < 0.05$). |

DeLacey et al. (2020), *PeerJ*, DOI 10.7717/peerj.8484

| Author, year | Dietary analysis | Micronutrient status | Clinical signs/symptoms and infections |
|---|---|---|---|
| *Panpanich et al. (1999)* | – | – | Illness in past four weeks (%) IBC: 35% FBC: 37% CC: 51% Undernutrition was present in 42% of IBC who had a history of illness in the last month compared with 18.8% of those who reported no illness ($p < 0.05$). |
| | | | HIV rates: IBC 23% (3/13) |
| | | Asia | |
| *Hearst et al. (2014)* | – | The nutritional status, based on blood biomarkers, revealed that 37.1% of the children were anemic, 21.4% had low albumin, 38.1% had low vitamin D, 5.5% were iodine-deficient and 2% had low serum zinc. | – |

DeLacey et al. (2020), *PeerJ*, DOI 10.7717/peerj.8484

**Table 3** (*continued*)

| Author, year | Dietary analysis | Micronutrient status | Clinical signs/symptoms and infections |
|---|---|---|---|
| *Kapavarapu et al. (2012)* | Dietary intake was compared with the Indian Recommended Dietary Allowance (RDA). A 24-h dietary recall revealed that children <7 years received 75% of the RDA for energy, and older children received 93 to 107% of RDA for energy. All children received adequate (>100% RDA) amounts of both protein and fat. | Hemoglobin (Hb) level was measured using automated blood analyzer. Results indicated that anemia was a prominent manifestation of HIV. Although baseline prevalence of anemia was only 40%, during the study period the cumulative incidence rose to 85%. | 75% had infections in the initial period (of <3 months) of admission into the facility. Pulmonary tuberculosis: 8% Impetigo: 31% Varicella zoster: 24% Chronic suppurative otitis media: 15% Parotitis: 13% HIV Group Home |
| *Kroupina et al. (2014)* | – | Venous blood samples were used for assessment of hemoglobin status. Anemia status was not found to be predictive of development status. "A significant percentage of the children in Kazakh institutions have micronutrient deficiencies; most strikingly, over half the sample was found to be anemic." | – |
| *Lewindon et al. (1997)* | – | – | Children with disabilities in long-term care at increased risk for H. pylori infection. 61% were seropositive for H. Pylori. 55.4% of 157 pediatric patients (<16yrs) were seropositive compared with 50 control group children ($p > 0.0002$). Children with disabilities frequently have excessive drool and contact with saliva could be an opportunity for the transmission of H. plyori. |

DeLacey et al. (2020), *PeerJ*, DOI 10.7717/peerj.8484

**Table 3** (*continued*)

| Author, year | Dietary analysis | Micronutrient status | Clinical signs/symptoms and infections |
|---|---|---|---|
| *Myint et al. (2012)* | – | – | Ocular manifestations: 5.1% Systemic comorbidities: 40% Chronic otitis media: 26.6% Pulmonary tuberculosis: 13.3% HIV Group Home |
| *Sarma et al. (1991)* | Dietary intake was compared with the Indian Council of Medical Research's (1984) recommended dietary allowance (RDA). 1,150 children were selected for dietary analysis. Energy intakes fell short compared to the RDA for most children and the deficit was higher in older children when compared to younger children. | Most common nutritional deficiencies encountered: vitamin A (2–8.5%), vitamin B complex and anemia. | Pallor indicating anemia: 2–17% Phyrnoderma: 1.2–6.8% Angular stomatitis: 1–32% Dental mottling: 1–18% Dental decay: 1–22% Cough, cold, fever, diarrhea, infections of the skin, eyes and ear/nose/throat complaints were most common. Deficiencies and morbidities were more common in younger age groups. |

DeLacey et al. (2020), *PeerJ*, DOI 10.7717/peerj.8484

**Table 3** (*continued*)

| Author, year | Dietary analysis | Micronutrient status | Clinical signs/symptoms and infections |
|---|---|---|---|
| *Zahid & Karim (2013)* | Food intake was obtained by 24 h food-weighing method for seven days. The average food intake were calculated by using the Institute of Nutrition and Food Science. Total food intake was about double the intake of similar children in the 1995–96 nutrition survey. Mean energy (2,270 kcal), protein (65 grams), carbohydrate (335 grams) and fat intake (73 grams). Carbohydrates, protein and fat provide 59%, 12% and 29% of total calories respectively. Protein intake was 65 grams, about 50% higher than the requirement and the 1995–96 nutrition survey of the urban location of the same group. Energy intake was found 20% higher than requirement and about 42% higher compared to 1995–96 nutrition survey. Average intake of IBC was higher than the national intake and the nutritional status of IBC was also found to be better than the national average by any nutritional criteria. Studies consider this to be potentially attributed to better health and care system prevailing in the orphanage apart provision of high-calorie and protein-rich food and that the nutritional status IBC, who are nutritionally disadvantageous, can be improved through organized feeding and better hygienic conditions. | Mean intake calcium 826 mg, iron 31 mg, vitamin A 6,462 IU, carotene 10,508 µg, vitamin B1 1.60 mg, vitamin B2 1.64 mg, niacin 19 mg, vitamin C 111 mg and zinc 10.2 mg. "Compared to 1995–96 nutrition survey, IBC had significantly higher micronutrient intake." | – |

**Table 3** (*continued*)

| Author, year | Dietary analysis | Micronutrient status | Clinical signs/symptoms and infections |
|---|---|---|---|
| | | Eastern Europe | |
| *Lesho et al. (2002)* | – | 90% of children had anemia and one-fourth had severe anemia. | 76% of children had parasites and 10% were infected with three or more. Disease frequency: Dermatologic: 17% Respiratory: 5% Genitourinary disorders: 3% Ear, nose and throat: 4% Psychiatric: 3% |
| | | European Union | |
| *Pysz, Leszczynska & Kopec (2015)* | Diets were chemically analyzed using the Kjeldahl method and Soxhlet method and compared to Polish Estimated Average Requirements. Results indicate that daily diets meet about 80% of recommended intake of energy, fat and carbohydrates. The intake of protein with daily diets exceeded EAR value and ranged from 115 to 362% (average 214.2%). It has been also found that the intake of basic nutrients was varied, coefficient variation (CV) ranged from 22.2% to 27.1%. Boys, compared to girls, spent almost twice as much time on physical activity. | – | – |

**Table 3** (*continued*)

| Author, year | Dietary analysis | Micronutrient status | Clinical signs/symptoms and infections |
|---|---|---|---|
| | | Middle East | |
| *El-Kassas & Ziade (2017)* | Compared to the Dietary Guidelines for American Children and Adolescents 2015 and based on a semi-quantitative food frequency questionnaire, more than half were estimated to have inadequate daily intake of vegetables, fruit, and proteins compared to the recommendation. 94.8% consumed three meals per day. 20.5% of adolescents ($\geq$10 years) reported meals did not satisfy appetite, compared to only 13% of children below 10 years, with no statistical significance between the two groups ($p = 0.480$). 45.1% of the studied sample revealed consumption of one snack per day; 49% consumed sweet and 19% consumed salty snacks on a regular basis. 82% of both age groups reported regular intake of breakfast. Inadequate protein intake (OR: 0.017, 95% CI [0.001–0.291]) was associated with statistically significant lower odds for being overweight and obese. Conversely, consumption of sweet snacks (OR: 6.492, 95% CI [1.124–37.512]) was associated with significantly higher odds for overweight and obesity. | – | Abnormal Hair Condition: 5.9% ($p = 0.736$) Abnormal Skin Condition: 26.1% ($p = 0.063$) Muscle Wasting: 2.6% ($p = 0.348$) Edema: 0% Bowing of legs or knocked knees: 2.6% ($p = 0.622$) Abnormal Mucus Membranes: 5.9% ($p = 0.014$) "Physical signs suggesting nutritional deficiencies were detected in about 25% of the sample." |

DeLacey et al. (2020), *PeerJ*, DOI 10.7717/peerj.8484

**Table 3** (*continued*)

| Author, year | Dietary analysis | Micronutrient status | Clinical signs/symptoms and infections |
|---|---|---|---|
| | | South America | |
| *Nunes et al. (1999)* | – | Anemia: 3% | "High rates of infectious diseases in all the children." HIV: 9.4% Gastroesophageal reflux: 3% Parasites: 6% Skin infections: 10% Upper respiratory infection: 7.3% Conjunctivitis: 1.7% |
| | | The Caribbean | |
| *Nelson (2016)* | Children living in both residential settings listed (1) carbohydrates and starches, (2) meat and (3) fruits and vegetables as the most commonly consumed food items. Significant difference in self-reports of foods consumed most often by CC and IBC (X2 (4, $N = 215$) =21.93, $P > 0.000$). CC were more likely than IBC to report meat as most often consumed food. Chi-square analyses revealed no significant differences in self-reports of foods consumed most often by IBC in the three orphanages. No significant differences in the food served at the three orphanages. No difference between physical activity between IBC and CC or between orphanages. | – | – |

**Notes.**

Study population: IBC, Institution-based care; FBC, Family-based care (orphaned or abandoned children in community settings); CC, Community children (non-orphans); CLS, Children living on the street.

live in IBC and there are a multitude of factors and reasons why they may be affected by different types of malnutrition. The extent and direction of this has not been well studied nor is it currently being effectively monitored or assessed (*Petrowski, Cappa & Gross, 2017*). Representing an inherently high-risk population, there are many reasons why we would expect undernutrition to be common: this was indeed observed in our review. Conversely, there are some reasons why IBC may offer opportunities for good nutrition and access to services, such as better food security, more reliable funding sources and access to specialized therapy or treatment. These ideal factors may not be possible or available for families affected by different economic circumstances living in the same communities (*Braitstein et al., 2013*; *Panpanich et al., 1999*; *Whetten et al., 2014*).

Our review, which used a comprehensive search strategy, also notably highlights a lack of well reported and standardized evidence. Only 19 countries were represented in our findings, despite Petrowski and colleagues finding 140 countries with data on children in institutions and this limited our ability to determine trends or region-specific patterns and risk factors (*Petrowski, Cappa & Gross, 2017*).

## Children with disabilities and children with low birth weight

A key observation is that few studies mentioned children with disabilities and only one included anthropometric analysis (Tables 1 and 2) (*Lewindon et al., 1997*). Children with disabilities are disproportionately present in institutionalized care settings. (*Baron, Baron & Spencer, 2001*; *The Children's Health Care Collaborative Study Group, 1994*; *The St Petersburg-USA Orphanage Research Team, 2005*). They are already at increased risk when they enter care centers because disabilities can increase the likelihood of being malnourished due to feeding challenges, malabsorption and/or intake needs. In addition, children with disabilities face the risk of their disabilities worsening in environments that do not meet their individual needs (*Groce et al., 2014*; *Kroupina et al., 2014*). Children with some types of disabilities may have higher caloric needs or require specialized diets or additional supports at mealtimes (*Groce et al., 2014*; *Johnson & Gunnar, 2011*; *Johnson et al., 2010*; *Kroupina et al., 2014*; *The Children's Health Care Collaborative Study Group, 1994*; *The St. Petersburg- USA Orphanage Research Team, 2008*).

Children in care or those who stay in care the longest may have more disabilities, more underlying diseases or more complex backgrounds—including a history of low birth weight (LBW), and therefore may require more focused care (*The Children's Health Care Collaborative Study Group, 1994*; *The St Petersburg-USA Orphanage Research Team, 2005*). Even when provided with adequate diet and medical care, these groups may be more dependent on caregivers for feeding, or need specialized approaches to feeding such as supportive seating and positioning, adaptive skill development and an extended time to eat (*Johnson & Gunnar, 2011*). When children enter into care they are often in poor health and those who stay the longest, such as some children with disabilities, are frequently in worse condition compared to children who are healthy at admission (*Groce et al., 2014*; *The St Petersburg-USA Orphanage Research Team, 2005*). These issues are important to highlight because becoming malnourished while living in an institution can also increase the risk of children developing a disability (*Groce et al., 2014*).

High prevalences of low birth weight infants were common within institutions; although child history, records or tracking were often limited (*Johnson et al., 2010*; *Kroupina et al., 2014*; *The Children's Health Care Collaborative Study Group, 1994*; *The St Petersburg-USA Orphanage Research Team, 2005*). Health status at birth was found to be a significant determinant of development. Growth trajectories and pre- and perinatal circumstances influence children's development in care: nutrition needs vary depending on individual growth rates and the presence of preexisting nutrition deficiencies (*Johnson et al., 2010*; *Kroupina et al., 2014*; *Martins et al., 2013*). *Johnson & Gunnar (2011)* and *Johnson et al. (2010)* found that during early rapid-growth phases, the effects of even modest nutritional deficits can become magnified. Age, age at admission and length of stay were other key factors identified that were associated with nutritional status (*Chowdhury et al., 2017*; *Kroupina et al., 2014*; *Martins et al., 2013*; *Panpanich et al., 1999*).

## Gender and malnutrition

Gender is also important to consider because programs and policies should be evidence-based and equitable, offering support to those most in need (*Theobald et al., 2017*). However, our review found that only nine of the studies compared genders. Of these, two found that girls were more malnourished, three found boys were more malnourished than girls and another four found both groups had similarly high prevalence of malnutrition or no significant difference in nutritional status by gender. We thus have mixed and inconclusive evidence of malnutrition or risk of malnutrition being linked to gender of children in institutional care (Table 2).  This may be a very context specific issue where social as well as biological factors play a role.

## Anthropometrics

Frequently the prevalence of low birth weight, stunting, wasting, underweight, anemia, and overweight was higher in IBC compared to the global prevalence for children younger than 5 years old (*The World Bank Group, 2019*). Paralleling global trends, the triple burden of malnutrition (undernutrition, micronutrient deficiencies and overweight/ obesity) also needs to be examined in IBC (*Black et al., 2013*; *UNICEF, 2019*). Although only a few studies reported on overweight, when it was reported, the prevalence was high, especially for adolescents. Future studies should report on overweight as well as underweight and micronutrient deficiencies. A positive feature of the studies reviewed was that many had peer groups for comparison; this is helpful because many children in the surrounding community may also deviate from WHO growth standards and it is helpful to see the nutritional status of children in IBC in local as well as global context. Multiple studies found that children in IBC were more undernourished than community children (CC) or children living in family-based care (FBC) (Table 2). Six studies indicated that peers within the community were more likely to be malnourished than children living within IBC, although this varied a bit by age. This could be in part due to children in care receiving adequate nutrition, routine meals and health screenings, especially for children who have HIV, and/or it could reflect the challenges faced by families in those communities (*Braitstein et al., 2013*; *Panpanich et al., 1999*; *Sarma et al., 1991*; *Whetten et al., 2014*; *Whetten et al., 2009*; *Zahid & Karim, 2013*).

### Clinical signs/symptoms, micronutrient status and infections

HIV prevalence was higher than global percentages for the few sites that reported it (*The World Bank Group, 2019*). HIV can be a significant risk factor for becoming malnourished and is also a contributing factor to children ending up in care (*Kotler, 1989*; *Leyenaar, 2005*). Another clear gap was that less than a third of the studies reported on micronutrient status and less than half reported on clinical signs/ symptoms or infections (Table 3). Micronutrient deficiencies were common with a prevalence of anemia higher than the global average in the majority of studies (*The World Bank Group, 2019*). The prevalence of micronutrient deficiencies in children in IBC is likely linked to their increased risk of sickness or morbidities (*Black et al., 2013*). *Hearst et al. (2014)* concluded that the growth-related indicators coincide with the high prevalence of low albumin, indicating generalized chronic undernutrition, and suggested macronutrient deficiencies could be due to inadequate diets, infections and/or inflammation, or impaired nutrient absorption or utilization secondary to the psychosocial stress of living in an institution.

### Dietary diversity, intake and food security

Only eight (32%) studies included information on dietary intake and, of those, half found intake or diet diversity to be inadequate. Dietary diversity was reported to be low for children in IBC, especially in terms of fruits, vegetables and protein. Limited funding and reliance on donations for food were frequently mentioned issues, and resulted in diets high in starches and legumes (*Mwaniki, Makokha & Muttunga, 2014*). Dietary adequacy varied; in some IBC sites children received an adequate amount or more than recommended dietary allowances and in others they received below the recommendations. Interestingly, the one study which reported on food security found that children in a Kenyan orphanage had higher food security when compared to children in FBC (*Braitstein et al., 2013*). However, it is impossible to generalize from this one study to say anything more broadly about food security.

### Limitations

We focused on nutritional status of children living in care but note that many other issues (e.g., development, cognition, puberty, catch-up growth, care practices, length of stay, age at admission, cause of institutionalization, illnesses, health of children who have been adopted or cultural practices) affect the demographics, health and well-being of children who are in institutions. It could be that all children coming into care are at risk due to the adverse events and trauma of being abandoned or orphaned (*Baron, Baron & Spencer, 2001*; *Martins et al., 2013*; *The Children's Health Care Collaborative Study Group, 1994*; *The St. Petersburg- USA Orphanage Research Team, 2008*). These wider factors were beyond the scope of this study (as well as infrequently reported in sufficient details in papers). Given biological links between poor nutrition and sub-optimal child development, evaluating these topics in more depth is critical in future work.

Although we found some research, there was limited recent information on this population of children. This may be because of practical or ethical considerations or it may reflect the desire to move away from institution-based care to family-based living

situations for children (*Kelley et al., 2016*). This review also only analyzed data from research published in English from January 1990 to January 2019. The studies were of differing designs and types. The review did not find enough studies to be able to examine differences between IBC, FBC, CC and CLS (children living on the streets). Other weaknesses included the common use of non-standard reporting methods or lack of clarity around measurement methods, such as how studies assessed micronutrient status or clinical signs and symptoms or determined disability status. Many of the studies were examining other subjects and nutritional/anthropometric information was only supplementary. Furthermore, growth measurements may have been affected by measurement or other errors (e.g., incorrect birthdate estimates leading to incorrect $z$-score calculations for age-related indices). Additionally, children with some types of disabilities may be shorter or lighter not because of inadequacy of dietary intake but because of their specific underlying conditions (e.g., disabilities such as Down syndrome and many others are associated with non-standard growth and development). It is also possible that there is under-diagnosis or misdiagnosis of medical conditions, chronic diseases or disabilities in these settings, which can also impair the growth and development of children (*Byass, Kahn & Ivarsson, 2011*). Another consideration is the potential for healthy survivor bias and sampling bias: some of the most vulnerable children may have died prior to measurement; younger children and healthier children may more quickly leave institutions with the remaining older residents more likely to have deficiencies (*van IJzendoorn et al., 2011*; *The St Petersburg-USA Orphanage Research Team, 2005*).

Risk of bias was apparent in most of the studies. We had originally considered using a formal risk of bias tool to differentiate study quality but did not do so because it became apparent that all of the studies had a high risk of bias and could not be representative of all the institutions in the countries. Another concern was that many used convenience sampling. It is also plausible that the sites included in the research were better-off facilities, which welcomed researchers, who were looking to share positive results and good performance. These are unlikely to be representative of all sites; we speculate that the overall situation is likely worse at many facilities with higher prevalence of malnutrition indicators. There is also wide variation between different institutional care facilities (*van IJzendoorn et al., 2011*; *Petrowski, Cappa & Gross, 2017*; *Whetten et al., 2014*).

## CONCLUSIONS

A key finding from this study was the limited amount of quality evidence-based data available on the nutritional status of children in institutions. Equally as important, our review found that where data was available, children living in institutionalized care were consistently at high risk of malnutrition, commonly experiencing undernutrition, overweight and/or micronutrient deficiencies. The implications for caregivers, clinicians, institutional administration and policy makers is that work is needed to ensure all children's basic rights to nutrition are met. Children living within care are at risk and require special attention. This is especially true for children with disabilities and low birth weight infants.

Although institutionalized care is not the ideal setting for children to grow up in, living within care continues to be a reality for many children. This study is in agreement with

other papers and reports that support optimizing current institutional environments when alternative placements for orphaned or abandoned children are not available. These children have a right to good nutrition, both to maintain their health now and to allow them to grow into healthy adults. Interventions will need to be multifaceted to address all of the root causes of malnutrition faced by children living in care. The need for much more evidence as well as a commitment to monitoring and evaluation of nutritional status in all institutions, should be acknowledged and children supported through improved nutrition programming as part of broader policy and child rights initiatives.

### Funding

The authors received no funding for this work.

### Competing Interests

Emily DeLacey, Michael Quiring and Caryl Garcia are employed by Holt International.

### Author Contributions

- Emily DeLacey and Marko Kerac conceived and designed the experiments, performed the experiments, analyzed the data, prepared figures and/or tables, authored or reviewed drafts of the paper, and approved the final draft.
- Cally Tann conceived and designed the experiments, analyzed the data, authored or reviewed drafts of the paper, and approved the final draft.
- Nora Groce, Maria Kett, Michael Quiring, Ethan Bergman and Caryl Garcia analyzed the data, authored or reviewed drafts of the paper, and approved the final draft.

### Data Availability

The raw data in available in Tables 1–3.

### Supplemental Information

Supplemental information for this article can be found online at http://dx.doi.org/10.7717/peerj.8484#supplemental-information.

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
