# Peer review of "The nutritional status of children living within institutionalized care: a systematic review"

_PeerJ, doi:10.7717/peerj.8484_

## Round 0.1 · original submission · Major Revisions

Dear authors,

Based on the comments of the external reviewers, your paper needs some corrections before being considered for publication in PeerJ. Please, see the comments below so as to have more information.

Best regards,
Dr Palazón-Bru

Reviewer 1 ·

Basic reporting

In general, it is an interesting article with a very specific topic. It is generally well-organized, but I think the discussion should be revised (see specific comments)

Experimental design

It is clear and the tables are very detailed.

Validity of the findings

More discussion about study limitations and some specific aspects about the micronutrients deficiencies and, importantly, the most common clinical problems in institutionalized children should be provided (see specific comments)

Additional comments

Introduction:
- paragraph 3 (especially from line 97 to 103) could be shortened or removed, as it contains information that are not essential for the study and its discussion, which is focused on nutritional issues in institutionalized children.
- paragraph 4 (line 104 to 127) should be summarized, by retaining the main concepts and information that are functional for the interpretation of the results and the eventual discussion.

Materials and methods are described appropriately.

Results:
- in addition to describe the output of the literature search in the text, I would suggest adding a Figure showing the flow chart of the extraction and selection process of articles, according to the PRISMA guidelines.
- line 393: define CWD; in general, define abbreviations at their first appearance in the text.
- Overall, the results section is quite clear.

Discussion:
- the first paragraph is quite interesting, as it seems that over the time the malnutrition of younger children can be overcome, since older children have fewer issues under this point of view. If we exclude some kind of bias (that I would recommend the authors to consider and discuss anyway), the authors tried to provide an explanation, namely the fact the most malnourished children may have higher likelihood to die, basically. By the way, in this regard, I suggest rephrasing this sentence “poorly nourished young children may also not survive to become older children in institutional settings”, by using the term risk and supporting this statement with additional references (e.g. Arch Public Health. 2018 Jul 16;76:28. doi: 10.1186/s13690-018-0277-1.; Am J Clin Nutr. 2013 Apr;97(4):896-901. doi: 10.3945/ajcn.112.047639.). Finally, the statement “Diet inadequacy, micronutrient deficiencies and illnesses or infections were also found to be prevalent”: where? In younger children?
- second paragraph: at some points, the discussion is difficult to be read and there are several repetitions, without addressing the main issue. For instance, specify the “reasons why IBC may offer opportunities of good nutrition”. Moreover, the authors state “Our review also highlights several issues in this field of study, most notably a lack of well reported and standardized evidence”: which issues? Finally, “Despite Petrowski finding 140 countries with data on children in institution” should be rephrased.
- third paragraph: “Children with disabilities are disproportionately present in institutionalized care settings”…that may be another good reason to explain the greater prevalence of malnutrition in younger children, if we assume that a significant proportion may have neurological disabilities since birth. Can the authors retrieve data in this regard? If my assumption is correct, they may consider this observation in the previous discussion art paragraph one as well.
- paragraph #4: actually, this issue is partly discussed in this paragraph; therefore, I suggest the authors to go over and review the organization of the whole discussion, in order to follow a more linear logic line. Actually, clearly dividing and organizing the discussion in several section, as the authors did in the results and maybe following the same organization in the discussion as well, could be the best solution, in my opinion, in order to make the discussion clearer and easier for the reader.
- Similar observations can be made on all following paragraphs, in general.
- Additionally, I suggest the authors to consider additional paragraphs/section to discuss some specific health issues that occur frequently in institutionalized settings (e.g. diarrhea, respiratory infections, anemia). In this regard, the authors just mentioned also the concern about the medical care provided to these institutionalized children. Most papers come from developing countries: e.g. in Asia (which is the most represented country in this article, probably) common diseases, even not infectious (for instance, celiac disease, which is not rare, very heterogeneous clinically, but invariably associated with malnutrition and/or micronutrients deficiency) can be completely overlooked in healthy and normal settings and, then, the authors can figure out in institutionalized children (e.g. Medicina (Kaunas). 2019 Jan 12;55(1). pii: E11. doi: 10.3390/medicina55010011.; PLoS One. 2011;6(7):e22774. doi: 10.1371/journal.pone.0022774).
- the final part about the study limitations should be more concise, in my opinion, as well as the conclusion, whereby I would suggest avoiding references citations, since that should summarize the main findings and conclusions, already discussed in the previous sections.

Reviewer 2 ·

Basic reporting

It is really important paper given that so many children live in the context institutional care. It does a very good review of the existent literature. It has a great organization that allowed us to highlight the main areas such as: physical growth, disabilities, special health issues and nutrition. The discussion section provides us with main gaps and new directions.

Experimental design

very appropriate for the research question.

Validity of the findings

well identified topic and very focused on this topic analysis

Additional comments

Well written and very important paper to address challenges of sensitive children who live in institutional care.

·

Basic reporting

see general comments section

Experimental design

see general comments section

Validity of the findings

see general comments section

Additional comments

Reviewer comments
Title: The nutrition status of children living within institutionalized care: a systematic review
Reviewer: Jude Thaddeus Ssensamba, MPH, MSc, FHSM
Date: 30/09/2019

Introduction
I thank you for giving me the opportunity to review this manuscript. With malnutrition still a major global health threat, this review is timely more so for more vulnerable populations like children staying in institutionalized settings.
The manuscript “The nutrition status of children living within institutionalized care: a systematic review” looks at existing data to paint a picture of the burden of malnutrition among children staying in institutionalized care. The study looks at both extremes of malnutrition (Under and over nutrition), and macro and micronutrient deficiencies.
The authors went ahead to look at the burden of malnutrition-related factors like gender, disability, age, and associated co-morbidities like HIV, skin infections, eye conditions, respiratory tract infections, genitourinary disorders, among others.
I commend the authors for putting this manuscript together; however, to ensure a clear understanding for readers, the authors should/could further clarify some points mentioned below:

General comments
The authors could consider defining the limits of what they describe as a child with regard to age. They could use international standardization.

Abstract
1. Lines 42-47 in the results section seem to fit more in the methods section. The authors could consider effecting that change.
2. The authors could consider using more specific wording in the abstract and whole manuscript. Words like “a few exceptions (line 55)”, “younger children (line 55)” may need to be clearly defined
3. The authors could clarify the meaning and intent of the sentence: “High risk of bias was found” (line 57), and how this was ascertained.
4. Being that the study did not intend to assess the extent of availability of information on malnutrition among children in institutionalized care, the authors could consider rewriting the conclusion (lines 58-67) to reflect the burden of malnutrition first and then maybe comment on limited data availability

Introduction
1. Line 73, the authors could consider being more specific with regard to the “millions” of children affected. Let us have the absolute number.
2. I appreciate the authors for painting a global picture of the burden of malnutrition among children under 5years. However, being that this study focuses on all children in institutionalized care, many of whom are above five years, the authors could consider giving a global view of malnutrition with a focus on all children (0-15 years) as using the under-five age to create a case for all children (0-15 years) may raise issues.
3. Paragraph one (line 73-83) could be rephrased, and some sentences that may supposedly be hanging like line 81 may be changed to give it a good flow.
4. Lines 84-85 could be rephrased as “Orphans and children living in institutionalized care are at a higher risk of malnutrition,” and a fitting reference included.
5. Lines 91-95 could be rephrased to fit well with lines 84-90
6. For consistency, the authors could choose to use either “Institutionalised care” or “residential care” throughout the manuscript
7. The authors could add a reference to line 99
8. Line 102-103 “these are rights for all children…” seems to be hanging. The authors could consider to modify it or put it into context with the whole paragraph.
9. The authors may wish to be more specific with regard to the “numerous adversities” (line 104), and relate this well (sentence structure) with the whole paragraph.
10. The authors could consider being more specific with regard to the use of the word “further adversities” (line 115), and for lines 116-117, the wording “impacted physical,” and the differences between infections and diseases may need to be clarified to the reader.
11. Lines 124-126 could be rephrased and better punctuated to ensure proper flow

Methods
General comments
1. In view of the already existing information on malnutrition among children in institutionalized care (already conducted studies), the authors may need to explicitly highlight herein the purpose of this review. More so that it is already known that children in institutionalized care have a higher risk of malnutrition more so younger children, the disabled, and those that were of low birth weight at birth. Otherwise, it may look like a global level interrogation of the extent of existing evidence on malnutrition in institutionalized settings.
2. The methods section gives readers a detailed account of the “how” the study was conducted. For systematic reviews, we need to see how the data was aggregated, how the differences in study designs were reconciled, how the different measurements were compared, and any associated errors were accounted for. In line with this, authors may consider enriching their methods section to reflect this.

Other comments
1. The fact that the review analysed data from different study designs and types (line 141) should be included in the limitations section
2. The authors could include in the limitation section the fact that they only considered data for the period Jan 1990 to Jan 2019, given that other studies on malnutrition in institutionalized care had been conducted before 1990.
3. Lines 141-157 could be rephrased and may be broken into two paragraphs where one looks at the “study type, the period of interest and justification, and criteria for inclusion of studies,” and the other, the variables of interest “anthropometric indicators, micronutrient status, presence of comorbidities, etc and the justification.”
4. The tense used inline 148 “…these studies must address…” could be changed to reflect the past tense, and the whole paragraph rephrased to improve flow.
5. Line 153, the authors may consider changing the use of “or” since head circumference for age cannot replace mid-upper arm circumference
6. Line 158, could include a comma between “following which” and “full texts.”
7. Line 159, Could replace “us” with “a.”
8. Line 160, Include an “s” to discussion
9. Line 164, In the methods section, the authors could be more explicit on the extent and meaning of “another measure.”
10. Lines 165-166, the authors may highlight the differences between “reported disease,” “illness” and “infection” and how these were treated and reported during analysis

Results
1. In the context of this study which is to investigate the burden of malnutrition among children in institutionalized care, lines 170-189 seem to explain the “how” the sample of interest, geographic distribution, types of studies, were ascertained. And this seems to fit more in the methods section. The authors may consider shifting this section to them methods section after improving its flow.
2. Lines 181-182, The authors included a phrase that one study had no ages. This raises questions since age is a crucial determinant for one being a child or not. Authors should clarify how they treated this paper that had no age, and how they grouped it or could consider removing it from the analysis.
3. The authors could clarify the difference between lines 182-184 and lines 187-188, or they could consider fusing them into one sentence
4. It is of great importance that the authors highlight the disability aspect of the study (lines 191-199). However, since the primary objectives were to determine the nutrition status, the authors could consider sharing this first (lines 200-208) before they go deeper into other aspects like disability and malnutrition, clinical signs, etc.
5. Lines 196-199 could benefit from rephrasing to make it easier for readers to understand.
6. Lines 200-208, the authors could consider enriching this section by adding how malnutrition differed by gender, geography, and age groups(under 5years, 5-14 years[school aged children], and 15-18years), with specific statistics, and if this was not done during analysis, the authors could consider rerunning the analysis.
7. Lines 210-211, the authors could consider changing the phrasing since HIV is not a clinical sign or symptom, and sections 210-212 could best suit the methods section.
8. Line 213-214, the authors could be more specific with regards to “the higher prevalence” with regard to both groups so that readers can estimate the variance in prevalence. With
9. Section 209-220 could speak more about what was the most prevalent signs/symptoms/diseases visa vie the number of studies that reported what
10. Lines 215-216 the authors could clarify the meaning of the sentence “conditions associated with nutrition status were also noteworthy” or otherwise consider making it more specific or delete it.
11. Line 223, the authors could include statistics to qualify this statement
12. Lines 223-224 The limited funding aspect could be taken to the discussion section and referenced accordingly
13. Section 221-233, could be rephrased to improve flow and coherence

Discussion

The authors highlighted key findings and their likely implications in line with existing literature. That said, the section could benefit from the following:
1. Generally, the authors could consider restructuring the discussion section to discuss: malnutrition and IBC in general, then IBC malnutrition and age, IBC malnutrition and Gender, IBC malnutrition and disability, IBC malnutrition and HIV, IBC malnutrition and co-infections and infestations, etc. These could be small subheadings. This will make the flow better and reduce the risk of repetitions.
2. Lines 237-238, the authors could consider changing the phrasing “of the 25 papers, all of them” to “all 25 reviewed studies…”
3. Lines 235-248 could be rephrased improve flow and coherence
4. Lines 247-248 could fit more in the result section. Perhaps the likely reasons for the deficiencies and illnesses could be discussed here
5. Line 249, the authors could provide a basis for saying this is the first systematic review (For example; based on a review of literature, to our knowledge…) putting in mind that many studies may have been conducted before January 1990.
6. Line 251, the authors may consider using more specific language instead of “many reasons.”
7. Lines 259-262 seem to suit more the limitations section; the authors could consider looking at it more closely
8. Section 249-262 may benefit from rephrasing to improve the flow
9. Line 263, the authors may consider being specific about the few studies by providing a statistic which may be a percentage.
10. The authors could clarify lines 264-266 to answer the “so what” question
11. Lines 269-270, a reference is needed for this assertion.
12. Authors could consider to clarify sentence lines 277-279, and give them a reference
13. The authors could consider revising sentence 283-284 “finally, becoming malnourished…), and to improve the flow of the whole paragraph.
14. Consider writing line 286 in reported speech
15. Line 288 could benefit from clarification of “which status” at birth.
16. The authors could consider revising lines 295-298 as them neem not to sync with lines 293-294
17. The construction of the paragraph that covers lines 299-306 could be revised as lines 299-300 seem not to flow well with subsequent lines.
18. For lines 307-309, the authors compared IBC malnutrition with global malnutrition among children younger than five years. Given that IBC covers different age groups comparing them with under-fives may need rethinking.
19. For line 309, the use of words like “it may be” could be qualified with a reference
20. Line 316, authors could consider qualifying the “six.”
21. Line 322-326 could be another paragraph that talks about malnutrition and HIV, and the likely “whys.”
22. Line 325, authors need to qualify “only eight.”
23. The authors could clarify what lines 328-332 intend to portray to the reader in relation to the subject matter as it seems hard to comprehend.
24. The authors could consider rephrasing line 340
25. The authors could include a reference for the sentence that covers line 340
26. The fact that the study only examined studies from 1990-2019 could be included as a limitation
27. The authors could explicitly include the fact that reviewed studies were of different designs, which would make comparisons difficult and how this was overcome.
28. The authors could consider including some strengths of the study

Conclusion
1. The authors could consider starting with the most important findings concerning malnutrition visa vie the extent of the presence of reviewed data, then would comment later on the scarcity of data.
2. References are rarely in conclusions more so since conclusions are based on the general impression of the study.
3. Line 381 the word “orphanages” seems new, the authors should clarify if orphanages also mean IBC settings/residential care and if yes, this needs to be defined in the introductory section

References
These were well written, and the authors are commended for this

Tables
Table 1:
The authors may need to clarify the rationale for the inclusion of some studies:
- The author could clarify why Lesho 2002 was included in the analysis, given they did not know the age of the study subjects.
- Pysz 2015 was included in the analysis though the age range was 7-20 years. The upper limit (20 years based on international standards) does not point to a child. How was this appropriated for? The authors could think of this.

Table 2: Okay

Table 3:
The inclusion of HIV rates as signs and symptoms (Braitstein 2013, Panpanich 1999, Nunnes 1999) could raise questions. The authors could consider including another column to specify “disease condition,” which could cater for specific diseases like Tuberculosis, HIV, etc.


After a thorough review of the manuscript, I thank the authors for bringing out the fact that malnutrition among children staying in IBC remains a challenge. The study highlighted the fact that few studies have been conducted in this area, an aspect that requires more rigorous studies. Given availed information, this manuscript would benefit from rigorous changes with regard to structure, flow, English, and content. I hope that the comments I raised herein will help the authors to make it better.

---

## Round 0.2 · Minor Revisions

Still pending some changes suggested by the reviewers.

Reviewer 1 ·

Basic reporting

The paper has been significantly improved and the authors addressed most of my observations and comments.

Experimental design

Systematic research criteria and parameters are appropriate.

Validity of the findings

No major concerns.

Additional comments

There is only one point that I recommend the authors to address, mention and shortly discuss/emphasize in the discussion, as suggested in the previous review. See below the specific point and rebuttal.

Review #1
- Additionally, I suggest the authors to consider additional paragraphs/section to discuss some specific health issues that occur frequently in institutionalized settings (e.g. diarrhea, respiratory infections, anemia). In this regard, the authors just mentioned also the concern about the medical care provided to these institutionalized children. Most papers come from developing countries: e.g. in Asia (which is the most represented country in this article, probably) common diseases, even not infectious (for instance, celiac disease, which is not rare, very heterogeneous clinically, but invariably associated with malnutrition and/or micronutrients deficiency) can be completely overlooked in healthy and normal settings and, then, the authors can figure out in institutionalized children (e.g. Medicina (Kaunas). 2019 Jan 12;55(1). pii: E11. doi: 10.3390/medicina55010011.; PLoS One. 2011;6(7):e22774. doi: 10.1371/journal.pone.0022774).

Authors' reply #1
- Thank you for sharing, we reviewed this article and added additional information, line 122 on page

Review #2 (current)
- I think the authors should add in the discussion a paragraph or a few sentences mentioning the possibility that many medical conditions without acute symptoms or with blunted clinical manifestations can be easily overlooked in these institutionalized setting and, in particular, in low-middle income countries, which represent the provenience of most paper considered in this systematic review. The authors may consider several non-communicable diseases, but one very good example might be celiac disease (as suggested), which is probably the most frequent chronic disease (often pauci-symptomatic, manifesting with growth problems mainly) that can impair the growth and nutritional status in children (including the reported micronutrients deficiency) all over the world. Both articles previously recommended can be very useful to mention, discuss and emphasize this specific aspect in the discussion. The additional information provided in the introduction (line 122, page 4) does not address this specific observation/suggestion appropriately, in my opinion. Indeed, table 2 (that is very detailed and informative) may indirectly suggest how chronic/no acute conditions are not considered in almost all available meta-analyzed studies. This issue of the under-diagnosis of medical conditions in settings with limited resources and/or poor awareness of specific non-acute diseases impairing the growth in children, should be mentioned.

·

Basic reporting

No comment

Experimental design

No comment

Validity of the findings

No comment

Additional comments

I would like to take this opportunity to thank the authors for the great work done to make this manuscript what it is now. All comments i raised have been adequately responded to.

With the exception of a few areas that can benefit from editing services, I recommend that this article passes on to the next publication phase.

I just included a few comments on the attached PDF that authors can respond to at their discretion.

---

## Round 0.3 · accepted · Accept

All the reviewers' concerns have been correctly addressed.

Reviewer 1 ·

Basic reporting

Clear and appropriate English Language.

Experimental design

Experimental design well defined

Validity of the findings

Results appropriately presented. The discussion might have been more improved and more focused, according to the previous suggestions.

Additional comments

No major concerns.